# Distribution Characteristics of Rainfall Erosivity in Jiangsu Coastal Areas

Feng Chen [1,2], Haibo Hu [1,*], Defeng Pan [3], Junyi Wang [2], Hua Zhang [2] and Zheng Pan [2]

1   Co-Innovation Center for the Sustainable Forestry in Southern China, Nanjing Forestry University, Nanjing 210037, China; chenfeng@njfu.edu.cn
2   Jiangsu Hydraulic Research Institute, Nanjing 210017, China; junyi9411@163.com (J.W.); zhanghuajssl@163.com (H.Z.); 211610010014@hhu.edu.cn (Z.P.)
3   Jiangsu Coastal Water Conservancy Research Institute, Yancheng 224200, China; pandefeng008@163.com
*   Correspondence: huhaibo@njfu.edu.cn; Tel.: +86-189-3600-6581

**Abstract:** The issue of regional soil and water loss caused by human activity is particularly severe in coastal regions. Since coastal reclamation areas are a valuable land reserve resource, it is of practical significance to understand the distribution characteristics of rainfall erosion and its impact on soil erosion for the prediction, evaluation, and management of regional soil and water resources. Rainfall erosivity should be updated and estimated from simplified indices. This paper analyzed the observed rainfall data of field runoff plots in Dongtai City, Jiangsu Province, between 2011 and 2017. According to the standard of erosive rainfall in coastal areas, reporting 10.8 mm of rainfall or 7.6 mm·h$^{-1}$ of $I_{30}$ (maximum 30 min rainfall intensity), the annual average erosive rainfall frequency in Dongtai City was 37.7 and the annual erosive rainfall was 1082.0 mm on average, which accounted for 51.6% and 90.6% of the total rainfall frequency and the total rainfall, respectively. Moreover, the annual average rainfall erosivity in the region from 2011 to 2017 was 7717.4 MJ·mm·hm$^{-2}$·h$^{-1}$. The annual distribution of rainfall erosivity was irregular, with an average monthly erosivity value of 4501.8 MJ·mm·hm$^{-2}$·h$^{-1}$. Since the accumulated rainfall erosivity of Dongtai City in the flood season (May to September) accounted for 88.1% of the total rainfall erosivity, it is essential to focus on preventing soil and water loss in the flood season. This paper established a rainfall-based model and a composite model and intensity appropriate for a single event and monthly rainfall erosivity in the region. Both models can be used to calculate the annual rainfall erosivity, but only the composite model based on rainfall amount and intensity is recommended for calculating single and monthly rainfall erosivity levels in Jiangsu coastal areas. The empirical formulas in Jiangsu coastal areas can be updated using more recent rainfall data and assess soil erosion risk accurately.

**Keywords:** erosive rainfall; soil erosion; rainfall intensity; coastal areas

## 1. Introduction

One of the major environmental catastrophes in the globe is the loss of soil and water. Extensive and in-depth observations and research on soil erosion and its impact factors have been conducted by numerous academics and specialists both domestically and internationally [1–5]. Soil erosion is the outcome of the combination of natural and social factors, and rainfall is the primary driving force leading to soil erosion among natural factors [6,7]. As one of the focuses of soil erosion research, rainfall erosivity reflects the impact of rainfall on soil loss and serves as a significant factor in the Universal Soil Loss Equation (USLE), Revised Universal Soil Loss Equation (RUSLE), and Chinese Soil Loss Equation (CSLE) models [8–12]. Rainfall is one of the main dynamic factors causing soil erosion. $EI_{30}$ is used to describe the potential capability of rainfall erosion quantitatively in the USLE and RUSLE models. It is complicated to calculate $EI_{30}$ and rainfall process data are needed. So, many simple methods of estimating rainfall erosivity have been studied, and rainfall erosivity has been evaluated and calculated using the conventional rainfall

statistical data of meteorological stations [13,14]. The erosion atlas used in the state of Hesse in Germany is still based on spatially interpolated rain gauge data and regression equations derived in the 1980s to estimate rainfall erosivity. Kreklow [15] showed that R-factors (rainfall erosivity factors) have increased significantly due to climate change and that current R-factor maps need to be updated using more recent and spatially distributed rainfall data.

Calculating the rainfall erosivity factor includes three aspects, single rainfall erosivity, period rainfall erosivity, and multi-year average annual rainfall erosivity [16]. The calculation models of rainfall erosivity can be classified into the single rainfall-based model and the composite index model. The single rainfall-based model is primarily based on daily, monthly, and annual rainfall data. The typical examples include a simple model developed by Richardson et al. [17] and Lee et al. [18] for estimating rainfall erosivity in the United States, South Korea, and other regions, as well as an algorithm proposed by Zhang et al. [19,20] that estimates semi-monthly rainfall erosivity based on daily precipitation. The composite index model takes rainfall amount and intensity into consideration, such as the yearly rainfall erosivity model established by Bu et al. based on the product of the flood season rainfall and the maximum 30 min rainfall intensity ($I_{30}$) [21]. Panagos et al. [22] presented the results of an extensive global data collection effort whereby they estimated rainfall erosivity for 3625 stations covering 63 countries. For the stations (located in China) the calculation of the rainfall erosivity, based on high temporal resolution data (1998–2012), was performed for the first time.

The erosive rainfall standard is a key index used to calculate soil erosion, which is of great significance for assessing multi-year average annual rainfall erosivity and the amount of soil erosion. Significant amounts of observation data indicate that only some rainfall events result in actual soil erosion, known as erosive rainfall [23–29].

Climate change has caused a large increase in R-factors (rainfall erosivity factors), which necessitates current R-factor maps to be updated with more recent and spatially distributed rainfall data [16,30]. The potential average increase in global rainfall erosivity was estimated to be 26.2~28.8% for 2050 and 27~34.3% for 2070 when compared to the baseline year of 2010 [31]. The rainfall erosivity factor (R) requires high-frequency data or can be estimated from simplified indices when the data are not available [32].

The coastline of the Jiangsu Province measures roughly 1071.2 km in length, and the water and soil resources have undergone dynamic development. It is located 1040 km from the Lianlu tidal flat in Dongtai City, which is on the Jiangsu Coast Area. Additionally, it silts due to upstream scouring and sediment flowing more than 100 m to the east annually, which is typical of a silted coast area [33]. Jiangsu's muddy coast was the site with the highest anthropogenic pressure and highest vulnerability [34]. Dongtai City is a typical location of a silted coast and is situated in the southern part of Jiangsu Province's coast. The erosive rainfall standard and rainfall erosivity in coastal areas were investigated in this study, using Dongtai City as an example. This paper calculated rainfall erosion in Dongtai City, Jiangsu Province, using the rainfall erosion formula in the RUSLE model, and discussed the erosive rainfall standard and interannual distribution characteristics in coastal areas based on the observed and rainfall data from 2011 to 2017. Moreover, the preliminary empirical formula for estimating rainfall erosivity and soil erosion modulus was proposed for its special environment characteristics (synoptic climate over the offshore, newly formed land reclaimed and silted in coastal areas) in this paper to provide a reference for predicting the amount of soil erosion in similar areas.

## 2. Materials and Methods

### 2.1. Overview of Test Area

The test area (2011–2017) is situated in Dongtai City ($32°33'$~$32°57'$ N, $120°07'$~$120°53'$ E), Jiangsu Province. With an average annual temperature of 15.0 °C, it lies in the transitional region between the northern subtropical zone and the warm temperate zone. In Dongtai, the prevailing direction of the wind is southeast and northwest, with an average yearly

wind speed of 3.3 m/s; there are 2231.9 h of sunshine on average each year. Based on meteorological statistics from 1956 to 2021, the average annual precipitation in the city is 1058.5 mm, with a maximum of 1978.2 mm (1991) and a minimum of 462.3 mm (1978); the average rainfall in the flood season (May to September) is 733.4 mm, with a maximum of 1294.1 mm and a minimum of 218.5 mm. Notably, the rainfall in flood and non-flood seasons is greatly different in Dongtai, making it easy to form drought and flood disasters.

Dongtai City in Jiangsu Province is a part of the sandy and silty soil region that runs along the province's northern shore. The field runoff plot is situated in Touzao Town, which is 20 km inland, as shown in Figure 1. The soil type is Solonchaks, with an occasional layer of salt on the surface. The soil bulk density is 1.21~1.31 g/cm$^3$, and the soil texture in the region is mainly sandy soil and sandy loam. There are 7 runoff plots in total (1°, 2°, 5°, 15°, 26.5°, 35°, and 45°), comprising 3 standard plots (1°, 2°, 5°, 5 m × 20 m) and 4 microplots (15° (4 m × 2.5 m), 26.5° (4 m × 2.5 m), 35° (2.5 m × 2.5 m), and 45° (3.5 m × 2.5 m)), which are all located on the Qingkan (the area left between the toe of the downstream slope and the borrow pit) on the north bank of the Dongtai River. According to the Test Specification of Soil and Water Conservation (SL419-2007) and the runoff plot test study [35–38], a wedge-shaped cement protective wall was erected around the runoff plot, with the lower part of the plot equipped with a runoff gathering pit, and the lower end of the pit connected to a collecting tank. The seven runoff plots were bare lands with no conservation measures.

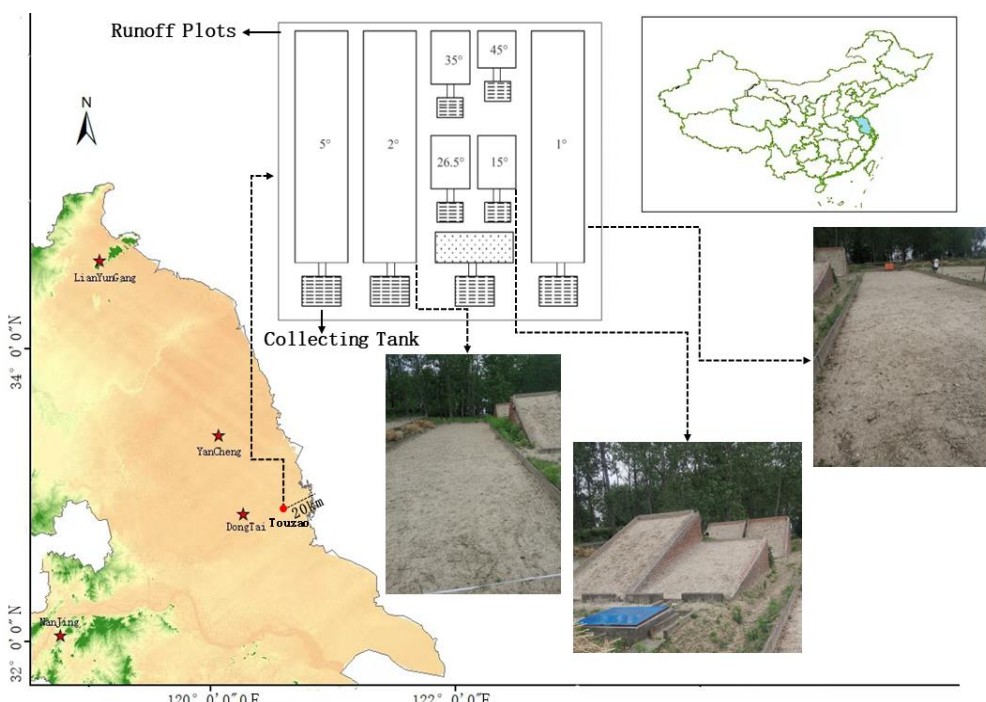

**Figure 1.** Layout and monitoring arrangement of the study area. The study area on the map of China. The study area on the map of Jiangsu. Seven runoff plots with gradients of 1°, 2°, 5°, 15°, 26.5°, 35°, and 45°. The stars and red dot represent place names and the experimental location.

*2.2. Data Sources*

From 2011 to 2017, observation experiments were carried out under natural rainfall conditions on the field runoff plots. The primary observation indices and techniques used in this study are as follows: a cutting ring method for measuring soil bulk density; a hydrometer method for soil structure particle gradation; a potassium dichromate method of external heating for measuring soil organic matter; a ProCheck handheld multi-function meter reading/data acquisition instrument for measuring soil salinity and verifying the 0~10 cm surface soil based on conductometry with a water–soil ratio of 5:1 (once a month); a soil moisture meter to measure soil moisture content once a week and to take additional measurements after rainfall; and a method to measure the runoff and sediment after rainfall-

runoff [39]. After each rainfall, the sand concentration of a water sample taken from the collecting tank was determined using the drying method, and the soil erosion amount was then calculated based on the water volume of the collecting tank and the area of the runoff plot. The meteorological data used in this paper (Dongtai station, 1956–2021) were obtained from the China Meteorological Data Service Center and the Dongtai Water Resources Information Network (www.dtslsw.cn, accessed on 12 December 2022).

*2.3. Analytical Methods*

2.3.1. RUSLE Model

As one of the widely utilized soil erosion prediction models at home and abroad, the RUSLE model was established based on USLE, a general soil loss model in the United States [10–12,40,41]. RUSLE is an empirical model whose parameters cover the main impact factors of soil erosion, as shown in Formula (1):

$$A = R \cdot K \cdot LS \cdot C \cdot P \tag{1}$$

where $A$ is the annual amount of soil erosion on average (t·hm$^{-2}$·a$^{-1}$); $R$ is the rainfall erosivity factor (MJ·mm·hm$^{-2}$·h$^{-1}$); $K$ is the soil erodibility factor (t·hm$^{-2}$·h·MJ$^{-1}$·mm$^{-1}$·hm$^{-2}$); $LS$ is the topographic factor (dimensionless); $L$ is the slope length factor; $S$ is the slope steepness factor; $C$ is the vegetation cover and management factor (dimensionless); and $P$ is the factor of soil and water conservation measures (dimensionless).

The rainfall erosivity factor $R$ adopts the classic index of calculation [41], as shown in Formulas (2)–(4):

$$R = E \cdot I_{30} \tag{2}$$

$$E = \sum_{r=1}^{n} (e_r \cdot P_r) \tag{3}$$

$$e_r = 0.29[1 - 0.72\exp(-0.05i_r)] \tag{4}$$

where $E$ is the kinetic energy of single rainfall (MJ·hm$^{-2}$); $e_r$ is the kinetic energy of rainfall per unit period (MJ·mm·hm$^{-2}$); $P_r$ is the period of rainfall corresponding to $e_r$ (mm); and $i_r$ refers to the rainfall intensity (mm·h$^{-1}$).

2.3.2. Rainfall Erosivity Calculation

As a dynamic metric for evaluating soil detachment and transportation levels brought on by rainfall, rainfall erosivity refers to the propensity for rainfall to induce soil erosion [7]. Zhang Wenbo and Fu Jinsheng [14] established simple erosivity algorithms based on different rain patterns, where the daily rainfall model had been widely used in the southern region with abundant rainfall due to its most stable performance and highest prediction precision [42]. Experts analyzed the changes in rainfall erosivity in the Yangtze River Basin, the Huaihe River Basin, the Pearl River Basin, and Guizhou [43–46]. Wang Yong [47] showed that the tillage erosion effects on soil and water loss were closely related to rainfall patterns in hilly agricultural landscapes.

The rainfall erosivity $R_c$ was calculated using Formulas (2)–(4) based on the observed data. The monthly rainfall erosivity $R_m$ and yearly rainfall erosivity $R_y$ were obtained by substituting the calculation results into Formulas (5) and (6).

$$R_m = \sum_{i=1}^{n} R_c \tag{5}$$

where $R_m$ refers to the monthly rainfall erosivity (MJ·mm·hm$^{-2}$·h$^{-1}$); $R_c$ refers to the single rainfall erosivity (MJ·mm·hm$^{-2}$·h$^{-1}$); and $n$ refers to the frequency of erosive rainfall in the month.

$$R_y = \sum_{i=1}^{12} R_m \tag{6}$$

where $R_y$ refers to the yearly rainfall erosivity (MJ·mm·hm$^{-2}$·h$^{-1}$); $R_m$ refers to the monthly rainfall erosivity (MJ·mm·hm$^{-2}$·h$^{-1}$); and $j$ refers to the month of erosive rainfall.

## 3. Results

### 3.1. Erosive Rainfall Standard

The erosive rainfall standard for the United States was determined to be 12.7 mm by Wischmeier based on test data from runoff plots around the country. The erosive rainfall standard in the Loess Plateau region was determined to be 10 mm, according to Jiang et al.'s [48] analysis of the rainfall data. However, the screening accuracy for evaluating rainfall events has not been analyzed. Wang et al. [24] examined the characteristics of erosive rainfall and soil erosion in different regions and proposed erosive rainfall standards appropriate for the regions. This paper provided statistics on the erosive rainfall standards (mainly in China) in some regions, as shown in Table 1, in accordance with the current relevant research.

**Table 1.** Erosive rainfall standard in some areas.

| No. | Researcher | Erosive Rainfall Standard (mm) | Proposed Time | Scope of Application |
|---|---|---|---|---|
| 1 | Wischmeier et al. [23] | 12.7 | 1978 | USLE/RUSLE model |
| 2 | Wang et al. [24] | 9.9 | 1984 | Loess Plateau in China |
| 3 | Yang Zisheng [49] | 9.2 | 1999 | Northeast mountainous area of Yunnan Province in China |
| 4 | Xie et al. [25] | 12 | 2000 | Loess Plateau in China |
| 5 | Cheng et al. [26] | 13.6 | 2004 | Southern mountainous area in Anhui Province in China |
| 6 | Ma et al. [50] | 11.2 | 2010 | Northern Jiangxi Province in China |
| 7 | Li et al. [27] | 11.3 | 2013 | Purple hilly area in China |
| 8 | Wang et al. [28] | 11.4 | 2013 | Red soil area in the north of Jiangxi Province in China |
| 9 | Wang et al. [7] | 8.47 | 2013 | Northern Shanxi Province in China |
| 10 | Bao et al. [29] | 4.45~21.15 | 2022 | Semi-arid Region of Northeast China |

The rainfall and measured erosion data of Dongtai City between 2015 and 2017 were compiled and analyzed in this paper. During the observation period, there were 258 rainfall events (rainfalls within 6 h are counted as the same rainfall event). Under ideal conditions, the sum of the rainfall erosivity of all rainfall events that meet the erosive rainfall standard equals the sum of the rainfall erosivity of all actual erosive rainfall events. According to this principle, rainfall erosivity was calculated using Formulas (2)–(4) in this paper based on the observed data and the rainfall erosivity of the actual rainfall event was added to obtain the sum of actual rainfall erosivity. All rainfall events in these three years were sorted from large to minor, and then the rainfall erosivity of the maximum rainfall event to the minimum rainfall event was sorted so that the accumulated value of rainfall erosivity was equal to the absolute value closest to the sum of the actual rainfall erosivity. At this point, the corresponding rainfall value of this rainfall event can be formulated as the rainfall standard of the erosive rainfall standard. The erosive rainfall standard in Dongtai City, a coastal area in Jiangsu Province, was concluded to be 10.8 mm of rainfall and 7.6 mm·h$^{-1}$ of I$_{30}$.

As shown in Table 1, the erosive rainfall standard in Jiangsu coastal areas was 10.8 mm, which was lower than the standard of 12.7 mm in the USLE/RUSLE model and lower than that of the mountainous areas in the southern Anhui Province and Jiangxi Province, which are situated in southern China as well. The fact that the erosive rainfall standard in Jiangsu coastal areas was relatively low across the country suggests that the area is prone to soil erosion.

*3.2. Distribution Characteristics of Erosive Rainfall*

3.2.1. Interannual Distribution Characteristics of Erosive Rainfall

The interannual distribution of erosive rainfall is an important influencing factor for the interannual distribution of soil erosion, which is of great significance for the further analysis of soil erosion [51]. Wu Yu-peng [13] compared nine methods to estimate the average annual rainfall erosivity levels using rainfall data from 3 to 16 years. Cardoso [30] analyzed rainfall erosivity for Pirassunga, SP, Brazil, using a 7-year rainfall dataset with a 10-min interval between measurements. The precipitation of Dongtai City in wet years ($p$ = 20%), normal years ($p$ = 50%), and dry years ($p$ = 75%) were 1271.4 mm, 1044.1 mm, and 882.9 mm, respectively (comprehensive planning of water resources in Dongtai City, 2012). There were three wet (1991, 1993, and 1998) years in 30 years (1970–2000), two wet (1998 and 2003) years in 14 years (1998–2012) that were used in the Global Rainfall Erosivity Database [22]. There are wet (2015 and 2016), normal (2011, 2012, and 2014), and dry (2013 and 2017) years in the observation period, making it an excellent example for research on the rainfall characteristics in the region. The erosive rainfall standard R of 10.8 mm or $I_{30}$ of 7.6 mm/h was utilized in this study to examine the rainfall data between 2011 and 2017. The outcomes illustrated 264 erosive rainfall events during the observation period, and the total rainfall amount was 7573.7 mm.

As shown in Table 2, the annual erosive rainfall events between 2011 and 2017 were at least 27 (2017) and at most 46 (2015), with a difference of 19 and an interannual coefficient of variation ($C_v$) of 0.16, and an average rainfall frequency was 37.7. The percentage of annual erosive rainfall in the total annual rainfall was 37.0–65.4%, with an average percentage of 51.6% and a $C_v$ of 0.17.

**Table 2.** Comparison between erosive rainfall and annual rainfall.

| Year | Erosive Rainfall Frequency | Annual Rainfall Frequency | $\frac{Erosive Rainfall Frequency}{Annual Rainfall Frequency}$ | Erosive Rainfall Amount (mm) | Annual Rainfall Amount (mm) | $\frac{Erosive Rainfall Amount}{Annual Rainfall Amount}$ |
|---|---|---|---|---|---|---|
| 2011 | 36 | 78 | 46.2% | 1143.5 | 1266.0 | 90.3% |
| 2012 | 38 | 73 | 52.1% | 919.0 | 1030.0 | 89.2% |
| 2013 | 34 | 52 | 65.4% | 722.5 | 774.0 | 93.3% |
| 2014 | 38 | 68 | 55.9% | 875.0 | 968.5 | 90.3% |
| 2015 | 46 | 79 | 58.2% | 1626.0 | 1748.5 | 93.0% |
| 2016 | 45 | 97 | 46.4% | 1657.6 | 1775.3 | 93.4% |
| 2017 | 27 | 73 | 37.0% | 630.1 | 747.2 | 84.3% |
| Average | 37.7 | 74.3 | 51.6% | 1082.0 | 1187.1 | 90.6% |
| Total | 264 | 520 | / | 7573.7 | 8309.5 | / |

The erosive rainfall amount between 2011 and 2017 was at least 722.5 mm (2013) and at most 1657.6 mm (2016), with an average of 1082.0 mm. The maximum erosive rainfall amount was 1027.5 mm (2.63 times) higher than the minimum, with a $C_v$ of 0.36. The percentage ranges of annual erosive rainfall amount in the total annual rainfall amount were 84.3% (2017) and ~93.4% (2016), with a $C_v$ of 0.03 and an average percentage of 90.6%. The erosive rainfall frequency between 2011 and 2017 was at least 27 (2017) and at most 46 (2015), with an average of 37.7. The maximum erosive rainfall frequency was 1.7 times the minimum, with a $C_v$ of 0.16. From the perspective of multi-year average data, 51.6% of erosive rainfall led to 90.6% of the erosive rainfall amount.

Table 2 demonstrates that although the variation trend of annual erosive rainfall was essentially consistent with that of annual rainfall, the interannual variation range of erosive rainfall was slightly greater than that of annual rainfall. The maximum rainfall amount was 1028.1 mm (2.38 times) higher than the minimum, with a $C_v$ of 0.33 (less than 0.36).

3.2.2. Intra-Annual Distribution Characteristics of Erosive Rainfall

The intra-annual distribution of erosive rainfall has a direct impact on the intra-annual distribution of soil erosion [5,51,52]. The observed data show that the average annual

erosive rainfall amount of Dongtai City was 1082.0 mm, accounting for 90.6% of the average annual amount of 1187.1 mm.

As shown in Table 3, erosive rainfall events occurred every month in the study area, with an average monthly frequency of 22 between 2011 and 2017. Overall, the rainfall was predominantly concentrated in the flood season (May to September), which made up 62.5% of the total frequency. The maximum frequency of rainfall events occurred in July, with 47, while the minimum occurred in January and February. Additionally, the cumulative erosive rainfall range was at least 118 mm in January and at most 1879.9 mm in July, with an average of 631.1 mm and a $C_v$ of 0.85. A large portion of erosive rainfalls was likewise concentrated during the flood season when the monthly totals exceeded 500 mm and made up 81% of the annual average. While erosive rainfall fluctuated significantly during the flood season, it was generally steady during the non-flood season. It is common for heavy rain to occur during the flood season due to the local climate, which is also the leading cause of soil erosion.

**Table 3.** Intra-annual distribution statistics of erosive rainfall (2011–2017).

| Month | January | February | March | April | May | June | July | August | September | October | November | December |
|---|---|---|---|---|---|---|---|---|---|---|---|---|
| Cumulative Rainfall Frequency | 9 | 9 | 11 | 20 | 27 | 24 | 47 | 44 | 23 | 14 | 23 | 13 |
| Average Rainfall Frequency | 1.29 | 1.29 | 1.57 | 2.86 | 3.86 | 3.43 | 6.71 | 6.29 | 3.29 | 2.00 | 3.29 | 1.86 |
| Cumulative Amount of Erosive Rainfall (mm) | 118.0 | 137.5 | 213.5 | 347.0 | 553.2 | 909.3 | 1879.9 | 1523.2 | 744.8 | 531.3 | 437.0 | 179.0 |
| Average Amount of Erosive Rainfall (mm) | 16.86 | 19.64 | 30.50 | 49.57 | 79.03 | 129.90 | 268.56 | 217.60 | 106.40 | 75.90 | 62.43 | 25.57 |

Table 3 demonstrates that the intra-annual variation in erosive rainfall was essentially constant and that the erosive rainfall was larger during the flood season than in the non-flood season. The intra-annual characteristic trend of erosive rainfall was unimodal, with a peak in July. The variation in erosive rainfall frequency had inflection points in June and October, and the rainfall frequency decreased more than the rainfall amount. It can be concluded that there were no consistent changes between erosive rainfall amount and its frequency by analyzing their variation tendency.

*3.3. Distribution Characteristics of Rainfall Erosivity*

3.3.1. Interannual Distribution Characteristics of Rainfall Erosivity

Annual rainfall erosivity is a key impact factor for soil erosion research and is of great significance in studying the spatiotemporal variation in soil erosion [43,44,51,53]. The observed data demonstrates that the yearly rainfall erosivity levels between 2015 and 2017 ranged from 2958.6 MJ·mm·hm$^{-2}$·h$^{-1}$ (2017) to 15,150.9 MJ·mm·hm$^{-2}$·h$^{-1}$ (2015), with an average of 7717.4 MJ·mm·hm$^{-2}$·h$^{-1}$ and a $C_v$ of 0.54. The maximum was 5.1 times the minimum, indicating that yearly rainfall erosivity varied and was discretely distributed. With the maximum 30 min rainfall intensity (83 mm/h) and the maximum rainfall erosivity (5876.8 MJ·mm·hm$^{-2}$·h$^{-1}$), the rainfall that started on 9 August 2015 led to the highest yearly rainfall erosivity. Short-term heavy rainfall and rapid surface runoff are the important factors causing serious soil and water loss on a short timescale in a mountainous region with yellow soil, which is of great significance for the construction of a regional soil erosion prediction model [54].

According to Figure 2, the interannual variation in rainfall erosivity and erosive rainfall between 2011 and 2017 was consistent, while rainfall erosivity's interannual $C_v$ of 0.54 was higher than 0.36 for erosive rainfall, indicating a larger variation range of rainfall erosivity. One possible explanation is that rainfall erosivity takes the impact of rainfall amount and intensity into account.

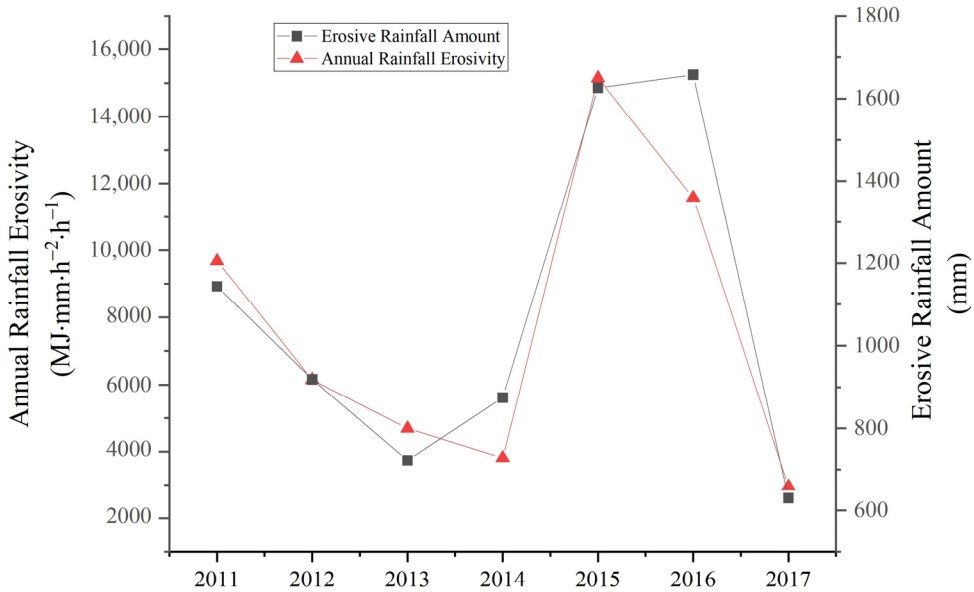

**Figure 2.** Annual distribution of rainfall erosivity and erosive rainfall amount.

### 3.3.2. Intra-Annual Distribution Characteristics of Rainfall Erosivity

It is crucial to examine the intra-annual distribution characteristics of the rainfall erosivity to forecast the amount of soil erosion in the test area since the intra-annual distribution of rainfall erosivity directly affects the intra-annual distribution of soil erosion [55,56]. The monthly rainfall erosivity between 2011 and 2017 ranged from 233.4 MJ·mm·hm$^{-2}$·h$^{-1}$ (February) to 18,326.4 MJ·mm·hm$^{-2}$·h$^{-1}$ (July), with an average of 4501.8 MJ·mm·hm$^{-2}$·h$^{-1}$ and a $C_v$ of 1.35, indicating an uneven intra-annual distribution of the monthly rainfall erosivity.

As shown in Figure 3, the intra-annual distribution curve of rainfall erosivity was unimodal, and its variation was similar to that of the erosive rainfall amount, with a peak in July. The flood season (May to September) had the highest rainfall erosivity, with a cumulative value of 47,602 MJ·mm·hm$^{-2}$·h$^{-1}$, accounting for 88.1% of the total rainfall erosivity. This percentage was higher than 81.1% of the cumulative erosive rainfall amount in the same period. The existence of the flood season (May to September) may explain why the percentage of rainfall erosivity is higher than that of the erosive rainfall amount in short-term rainstorms that frequently occur in summer, and the maximum 30 min rainfall intensity ($I_{30}$) was large. According to the observed data, monthly average rainfall erosivity during the flood season was 9520.4 MJ·mm·hm$^{-2}$·h$^{-1}$, compared to 917.1 MJ·mm·hm$^{-2}$·h$^{-1}$ in the non-flood season, indicating an enormous difference of 10.4 times. Therefore, the flood season (May to September) is the most important time for soil erosion prevention.

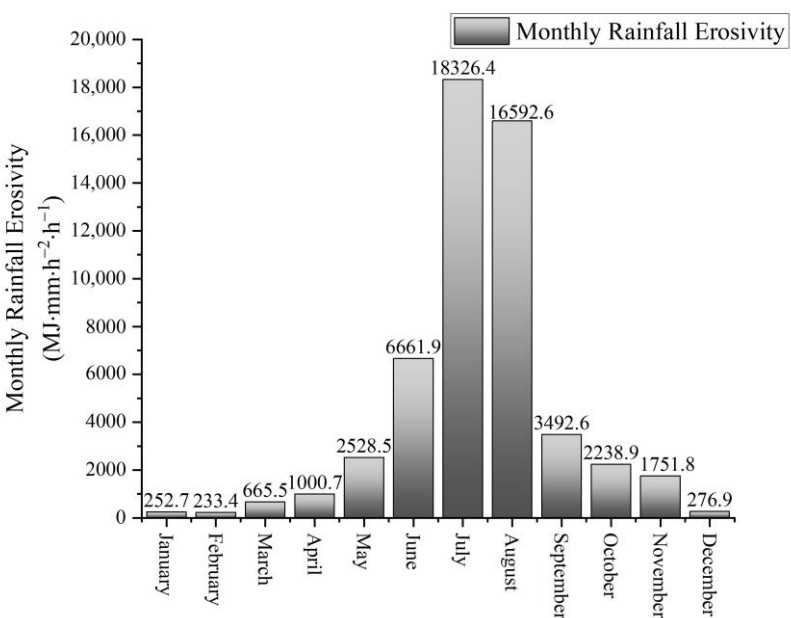

**Figure 3.** Intra-annual distribution of monthly rainfall erosivity (2011–2017).

## 4. Discussion

### 4.1. Empirical Formula for Rainfall Erosivity

A standard algorithm for rainfall erosivity indicators both at home and abroad was proposed by Wischmeier et al. using $EI_{30}$, the product of the total kinetic energy of rainfall, and the maximum 30 min rainfall intensity ($I_{30}$) as the index to quantify rainfall erosivity. A more precise and simple calculation method for rainfall erosivity is required since the computation of $EI_{30}$ requires precise rainfall data, and the data processing is quite time-consuming. Additionally, local rainfall characteristics, rainfall data acquisition, and the relationship between rainfall erosivity and $EI_{30}$ must all be considered when determining an algorithm for rainfall erosivity. Scholars at home and abroad have carried out extensive research on the algorithms [20,57], of which the two most representative models are the rainfall-based model and the composite model based on rainfall amount and intensity.

The rainfall-based model can be expressed with Formula (7) as follows:

$$R = aP^b \tag{7}$$

where $R$ is the rainfall erosivity (MJ·mm·hm$^{-2}$·h$^{-1}$); $P$ is the rainfall amount (mm); and $a$ and $b$ are coefficients.

The composite model based on rainfall amount and intensity developed by Foster et al., (1983). can be expressed by Formula (8) as follows:

$$R = cPI_{30} \tag{8}$$

where $R$ is the rainfall erosivity (MJ·mm·hm$^{-2}$·h$^{-1}$); $P$ is the rainfall amount (mm); $I_{30}$ is the maximum 30 min rainfall intensity (mm·h$^{-1}$); and $c$ is the coefficient.

Algorithms for the single, monthly, and yearly rainfall erosivity of the above two models were established based on the rainfall data between 2011 and 2017 through regression analysis, as shown in Table 4.

**Table 4.** Rainfall-based model and composite model based on rainfall amount and intensity.

| Model | Rainfall-Based Model and $R^2$ | Composite Model Based on Rainfall Amount and Intensity and $R^2$ |
| --- | --- | --- |
| Single Rainfall Erosivity Model | $R_c = 0.5309P_c^{1.5626}$ $(R^2 = 0.7542)$ | $R_c = 0.2314P_cI_{30}$ $(R^2 = 0.9793)$ |
| Monthly Rainfall Erosivity Model | $R_m = 0.2748P_m^{1.5821}$ $(R^2 = 0.8763)$ | $R_m = 0.1561P_mI_{30}$ $(R^2 = 0.9463)$ |
| Yearly Rainfall Erosivity Model | $R_y = 0.1792P_y^{1.519}$ $(R^2 = 0.89)$ | $R_y = 0.0976P_yI_{30}$ $(R^2 = 0.9031)$ |

Note: $R_c$ refers to the single rainfall erosivity level; $P_c$ refers to the amount of erosive rainfall; $R_m$ refers to the monthly rainfall erosivity level; $P_m$ is the amount of monthly erosive rainfall; $R_y$ refers to the yearly rainfall erosivity level; $P_y$ is the amount of yearly erosive rainfall; $I_{30}$ represents the corresponding maximum 30 min rainfall intensity of single, monthly, and yearly rainfall.

Table 4 demonstrates that, for the single, monthly, and yearly erosive rainfall models, the coefficients of efficiency of the composite model based on rainfall amount and intensity were greater than those of the rainfall-based model, indicating a higher prediction precision of the composite model compared to the rainfall-based model. Moreover, the coefficient of efficiency $R^2$ of the rainfall-based model rose as the timescale lengthened, while the $R^2$ of the composite model decreased gradually. In the yearly rainfall erosivity calculation, the $R^2$ values of the two models were close, both with a high goodness of fit of around 0.9, indicating that the rainfall-based model was applicable on a longer timescale. Therefore, it can be inferred that the composite model can be preferentially used when calculating the single and monthly rainfall erosivity levels of coastal areas, and the data obtained were of high accuracy. Both models were applicable for calculating yearly rainfall erosivity.

The empirical formulas of single, monthly, and yearly rainfall erosivity levels in Table 4 were compared with the rainfall erosivity in the RUSLE model in this study based on the rainfall data between 2018 and 2021, as displayed in Figure 4. It can be concluded that rainfall erosivity has a steady overall changing trend, as determined by the empirical formula and the RUSLE model. The RUSLE model's predicted value for rainfall erosivity is lower than the empirical formula's predicted value. Specifically, as for single rainfall, the calculated values of the rainfall-based model and the composite model based on rainfall amount and intensity were 37.62% and 18.54% higher than those calculated by the RUSLE model, respectively. Similarly, the calculated values of the monthly rainfall were 35.60% and 17.88% higher, and the values of the yearly rainfall were 56.35% and 30.34% higher, respectively. There was a significant difference between 2019 and 2020, with annual erosive rainfall values of 461.5 mm and 1005.5 mm, respectively. As for the comparison between 2018 and 2021, with an annual erosive rainfall of 779 mm and 930.5 mm, respectively, the calculated values of single, monthly, and yearly rainfall exhibited a difference of 2.62~12.84% between the RUSLE model and the composite model based on rainfall amount and intensity, and a difference of 14.10~32.89% between the rainfall-based model and the RUSLE model.

Rainfall erosivity exhibits a high spatiotemporal variability and has increased significantly due to climate change [14]. When estimating rainfall erosivity using conventional meteorological data, the criteria should be easy access to data, simple calculation, high accuracy, and regional stability [12]. According to an assessment of the empirical relationships for rainfall erosivity, the composite model is recommended for calculating single and monthly rainfall erosivity, and the rainfall-based model can be used for calculating annual rainfall erosivity in Jiangsu coastal areas. The empirical formulas for rainfall erosivity need to be tested and modified using more recent data to improve their accuracy and stability, and then they may be used to estimate current soil erosion rates or to estimate soil losses considering future climate change scenarios.

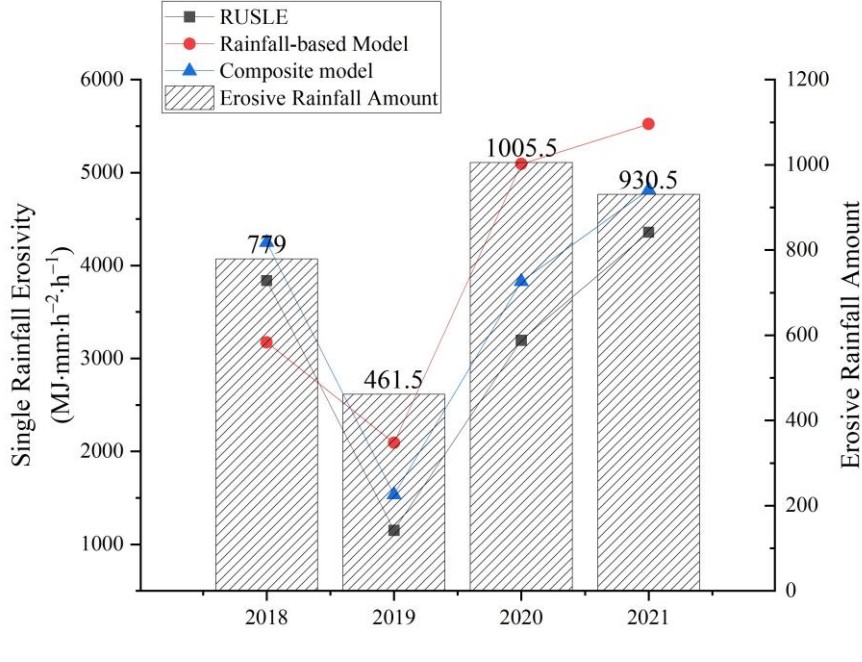

(**a**)

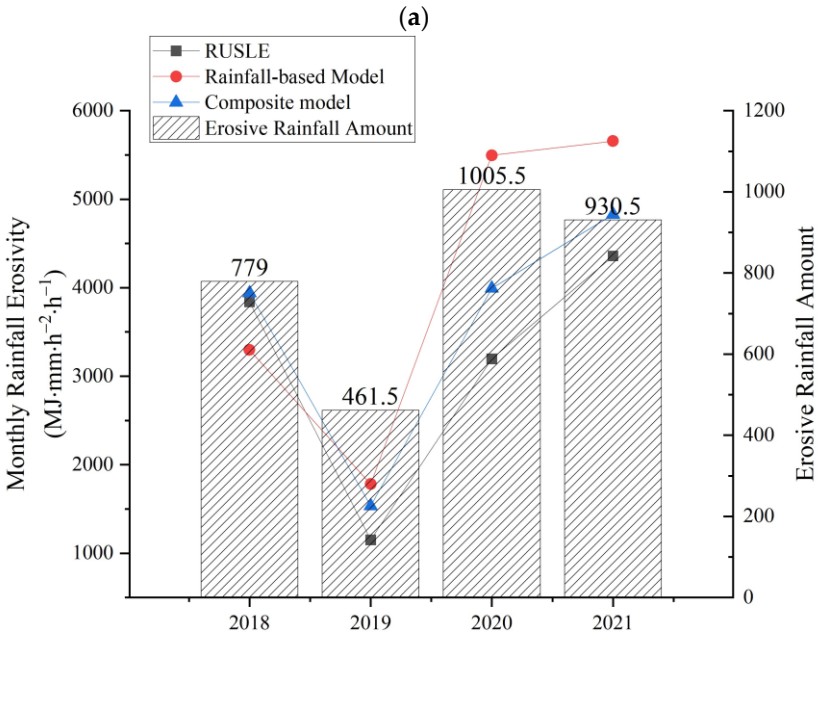

(**b**)

**Figure 4.** *Cont.*

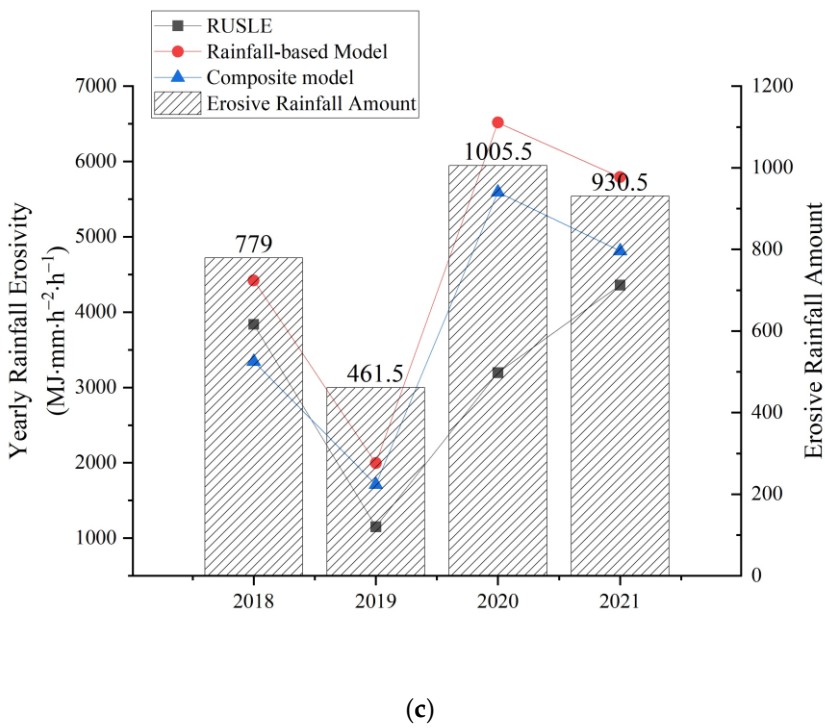

(**c**)

**Figure 4.** Value comparison between the rainfall-based model, composite model based on rainfall amount and intensity, and the RUSLE model. (**a**) Single rainfall erosivity model; (**b**) monthly rainfall erosivity model; (**c**) yearly rainfall erosivity model.

### 4.2. An Empirical Formula for Rainfall Erosivity and Soil Erosion Modulus

The rainfall erosivity index fully takes the impact of rainfall amount and intensity into account, which can more accurately depict the status of soil erosion theoretically. The soil erosion modulus in this study was thoroughly determined by the test plots with varied slopes (1°, 2°, 5°, 15°, 26.5°, 35°, and 45°), as the slope of newly formed land reclaimed and silted in coastal areas was not fixed. This study conducted regression analysis on rainfall erosivity and soil erosion modulus based on the data between 2011 and 2017 to further confirm the applicability of the rainfall erosivity index in the coastal areas of Jiangsu Province. The relationship between the two is depicted in Figure 5.

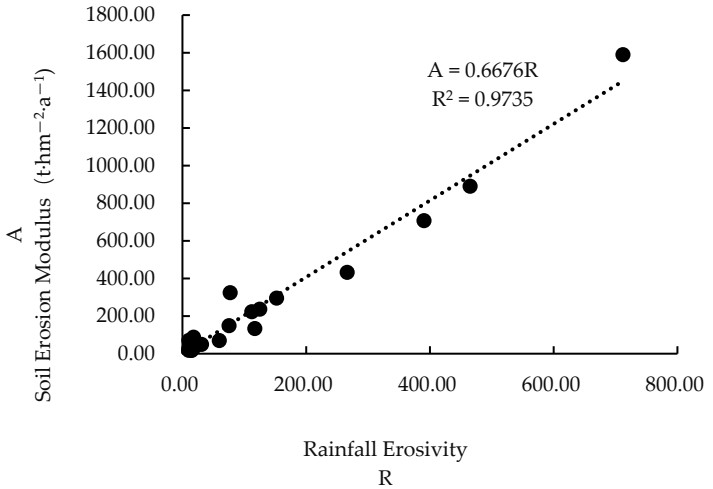

**Figure 5.** Relationship between rainfall erosivity and soil erosion modulus.

Figure 5 demonstrates that rainfall erosivity is positively correlated with soil erosion modulus. The relationship between the two can be expressed by an empirical formula through regression analysis, as shown in Formula (9):

$$A = 0.6676R \tag{9}$$

where $A$ refers to the soil erosion modulus ($t \cdot hm^{-2} \cdot a^{-1}$) and $R$ refers to the rainfall erosivity ($MJ \cdot mm \cdot hm^{-2} \cdot h^{-1}$). The effectiveness coefficient of this regression expression reached 0.9735. As a result, this expression can be utilized as an empirical formula for the soil erosion modulus in Jiangsu coastal areas to prevent soil and water loss. Based on this, the soil erosion modulus can be estimated by calculating rainfall erosivity.

## 5. Conclusions

In Jiangsu coastal areas with a valuable land reserve resource, regional soil, and water loss brought on by human activity is particularly severe.

The standard of erosive rainfall in Jiangsu coastal areas is 10.8 mm of rainfall or 7.6 mm/h of $I_{30}$ (maximum 30 min rainfall intensity).

The average erosive rainfall frequency in Dongtai City was 37.7 during the test period and the annual erosive rainfall was 1082.0 mm on average, which accounted for 51.6% and 90.6% of the total rainfall frequency and total rainfall, respectively.

The annual average rainfall erosivity value in the test region between 2011 and 2017 was 7717.4 $MJ \cdot mm \cdot hm^{-2} \cdot h^{-1}$. The annual distribution of rainfall erosivity was irregular, with an average monthly erosivity of 4501.8 $MJ \cdot mm \cdot hm^{-2} \cdot h^{-1}$. Moreover, since the accumulated rainfall erosivity of Dongtai City in the flood season (May to September) was 49,840.9 $MJ \cdot mm \cdot hm^{-2} \cdot h^{-1}$, accounting for 92.3% of the total rainfall erosivity, it is essential to focus on preventing soil and water loss in the flood season.

The empirical formulas for rainfall erosivity and soil erosion modulus were preliminarily established in this paper, which can be updated using more recent data and by assessing soil erosion risks in Jiangsu Province's coastal areas. The bias due to periods for calculating rainfall erosivity should be considered.

The situation of soil and water loss in Jiangsu coastal areas should be prioritized due to its impact on the surrounding environment, particularly during the flood season. Soil and water conservation methods should be developed on the exposed slope to facilitate soil erosion reduction.

**Author Contributions:** F.C.: funding acquisition, the acquisition of data, the analysis and interpretation of data, writing—original draft, and visualization. H.H.: writing—review and editing. D.P.: field experiment and supervision. J.W.: the acquisition of data, formal analysis, and validation. H.Z.: funding acquisition and writing—review and editing. Z.P.: writing—review and editing, and formal analysis. All authors have read and agreed to the published version of the manuscript.

**Funding:** This study was mainly supported by the Research Funds from the Jiangsu Hydraulic Research Institute (grant no. 2022Z019), the Jiangsu Science and Technology Program (grant no. 2020052), and the National Natural Science Foundation of China (grant no. 31400617).

**Data Availability Statement:** Data are contained within the article.

**Acknowledgments:** The authors would like to thank all funds and lab facilities. We also gratefully acknowledge the anonymous reviewers for their constructive comments.

**Conflicts of Interest:** The authors declare no conflict of interest.

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
