# Peer review of "Distribution Characteristics of Rainfall Erosivity in Jiangsu Coastal Areas"

_agronomy, doi:10.3390/agronomy13071829_

Round 1

Reviewer 1 Report

Dear colleagues, I have studied your paper.

His topic is certainly useful and will gain importance in the future.
I generally agree with the structure of the paper, but I have several suggestions for improvement.
The abstract is relatively general, especially in the first part. The information in it is not wrong, it would just like to make more concrete the novelty of the obtained data and its meaning.
The introduction of the paper is related to this. In principle, his information is correct and related to the paper, it would be good to supplement it and connect it more with the local significance and the reason for the research. It is indicated in the introduction (at its end), but it would be good to supplement this part.

The methodological part is basically correct, but it would be good to write more data about the location, in more detail about temperatures, precipitation and especially soil conditions. The methodology also describes basic models - especially RUSLE. Again, this is not wrong, however I think this is a well-known fact that a literary reference would suffice. However, if this part stays there, it doesn't matter.

The result part is simply and clearly described. However, I fundamentally lack a discussion preferably in a short chapter. I consider this to be an important reminder. The conclusion then corresponds to the paper.

Author Response

Dear Reviewer,

His topic is certainly useful and will gain importance in the future.

I generally agree with the structure of the paper, but I have several suggestions for improvement.

The abstract is relatively general, especially in the first part. The information in it is not wrong, it would just like to make more concrete the novelty of the obtained data and its meaning.

The introduction of the paper is related to this. In principle, his information is correct and related to the paper, it would be good to supplement it and connect it more with the local significance and the reason for the research. It is indicated in the introduction (at its end), but it would be good to supplement this part.

The methodological part is basically correct, but it would be good to write more data about the location, in more detail about temperatures, precipitation and especially soil conditions. The methodology also describes basic models - especially RUSLE. Again, this is not wrong, however I think this is a well-known fact that a literary reference would suffice. However, if this part stays there, it doesn't matter.

The result part is simply and clearly described. However, I fundamentally lack a discussion preferably in a short chapter. I consider this to be an important reminder. The conclusion then corresponds to the paper.

The suggestions are listed below:

Point 1: The abstract is relatively general, especially in the first part. The information in it is not wrong, it would just like to make more concrete the novelty of the obtained data and its meaning.

Response 1: The sentence of “Rainfall erosivity should be updated with more recent and spatially distributed rainfall data, and it can be estimated from simplified indices.” has been simplified in previous lines 14-15.

In order to make the language expression more explicit, the words of “serve as a guide for further study on rainfall erosivity in similar areas” in previous line 31 have been changed by “assess soil erosion risk accurately”.

Point 2: The introduction of the paper is related to this. In principle, his information is correct and related to the paper, it would be good to supplement it and connect it more with the local significance and the reason for the research. It is indicated in the introduction (at its end), but it would be good to supplement this part.

Response 2: We have further added the description of indicators related to the local significance and the reason for the research in previous lines 85-86. so that the whole introduction is more consistent with the research contents of this manuscript. In addition, we have supplemented the reference of [3] for the areas in previous line 86.

Point 3: The methodological part is basically correct, but it would be good to write more data about the location, in more detail about temperatures, precipitation and especially soil conditions. The methodology also describes basic models - especially RUSLE. Again, this is not wrong, however I think this is a well-known fact that a literary reference would suffice. However, if this part stays there, it doesn't matter.

Response 3: We have further revised the temperature and precipitation based on meteorological statistics from 1956 to 2021 in previous line 99 and line 104. The sentence of “The soil type is Solonchaks, with an occasional layer of salt on the surface.” has been added to describe the soil in previous lines 112-113.

For the convenience of later description, the description of RUSLE has been retained.

Point 4: The result part is simply and clearly described. However, I fundamentally lack a discussion preferably in a short chapter. I consider this to be an important reminder. The conclusion then corresponds to the paper.

Response 4: We have further supplemented a discussion for the recommended empirical formulas under different conditions and the application of empirical formulas in Section 4.1. At the same time, similar parts were removed from the conclusion (L459-464).

At the same time, we have read the manuscript many times and revised some expressions and writing errors. In addition, we have supplemented the reference of [51] for the tillage erosion effects in previous line 173 and [52] for the influence of short-term heavy rainfall in previous line 310. If you have any questions about this manuscript, please don’t hesitate to contact us.

Thanks again for your valuable comments and suggestions.

Best wishes

Feng Chen

Reviewer 2 Report

Dear Authors,

The subject of article is interesting, especially in time when climate is changing and many countries have the problem with growing wind, soil and water erosion. However, the work is prepared chaotically and requires thorough editing.

The manuscript is in general written in a clear way, easy to read and with appropriate reference to data in figures and tables, but Authors should add more information about the conditions of region. There is lack of the most important informations about weather conditions eg. temperature, precipitation at the analysed period. The text in many places are not understandable, what years the Authors analysed? The units should also be redrafted so that the reader can more easily understand the meaning of the work. In my opinion all text is not precise and needs correction.

All suggestions are included in the manuscript.

Best regards

Author Response

Dear Reviewer,

The subject of article is interesting, especially in time when climate is changing and many countries have the problem with growing wind, soil and water erosion. However, the work is prepared chaotically and requires thorough editing.

The manuscript is in general written in a clear way, easy to read and with appropriate reference to data in figures and tables, but Authors should add more information about the conditions of region. There is lack of the most important informations about weather conditions eg. temperature, precipitation at the analysed period. The text in many places are not understandable, what years the Authors analysed? The units should also be redrafted so that the reader can more easily understand the meaning of the work. In my opinion all text is not precise and needs correction.

All suggestions are included in the manuscript. They are listed below:

Point 1: Authors should add more information about the conditions of region. There is lack of the most important informations about weather conditions eg. temperature, precipitation at the analysed period.

Response 1: We have further revised the temperature and precipitation based on meteorological statistics from 1956 to 2021 in previous line 99 and line 104, and precipitations at the analysed period (2011-2017) were shown in Table 2.

Point 2: The text in many places are not understandable, what years the Authors analysed?

Response 2: This is our negligence. We have revised some expressions and writing errors according to the suggestions (see the manuscript).​ The analyzed period was seven years (2011-2017), and was added in previous line 98.

Point 3: The units should also be redrafted so that the reader can more easily understand the meaning of the work.

Response 3: We have redrafted the units according to the suggestions in the manuscript (L17, L145), including Figure 2, 3 and 4.

Point 4: remove it

Response 4: The words of “rainfall erosivity” in previous line 32 have been deleted. The sentence of “The following conclusions have been reached in this paper based on the analysis of the observed rainfall data from field runoff plots in Dongtai City, Jiangsu Province, between 2011 and 2017:” in previous lines 445-447 has been deleted. The words of “rainfall erosivity” in previous line 32 have been deleted. The words of “6. Patents” in previous line 473 have been deleted. The sentence of “This section is not mandatory but may be added if there are patents resulting from the work reported in this manuscript.” in previous lines 474-475 has been deleted.

Point 5: add the year

Response 5: We have added the publication year for “Richardson et al.” as “Richardson et al.(1983)” in previous line 61, the publication year for “Lee et al.” as “Lee et al. (2011)” in previous line 62 and the publication year for “Zhang et al.” as “Zhang et al. (2002)” in previous lines 63.

Point 6: “R-factors” explain what do you mean

Response 6: We have supplemented the mean of the words of “R-factors”. The words of “(rainfall erosivity factors)” have been added after the words of “R-factors” in previous line 74, and the same change in previous line 53.

Point 7: describe the soil according to WRB

Response 7: The sentence of “The soil type is Solonchaks, with an occasional layer of salt on the surface.” has been added to describe the soil in previous lines 112-113.

Point8: Modify the initial letter “F” to “f” in the word “Formula”

Response 8: We have modified according to the format of the journal.

Point 9: the table is not clear, please add the lines

Response 9: The lines have been added in Table 3.

Point 10: space

Response 10: The space has been added between the number “1082.0” and the unit “mm” in previous line 451.

Point 11: In my opinion all text is not precise and needs correction.

Response 11: This manuscript was previously submitted to MDPI for English editing with the following certificate.

At the same time, we have read the manuscript many times and revised some expressions and writing errors. In addition, we have supplemented the reference of [51] for the tillage erosion effects in previous line 173 and [52] for the influence of short-term heavy rainfall in previous line 310. If you have any questions about this manuscript, please don’t hesitate to contact us.

Thanks again for your valuable comments and suggestions.

Best wishes

Feng Chen

Reviewer 3 Report

This paper has presented the detailed analysis of rainfall characteristics in a study site in China where plot-based monitoring results were used to identify threshold value for erosive rainfall, quantification of their temporal and spatial variations and develop customised methodology for the estimation of rainfall erosivity. The topic covered (water and soil management on agricultural land) is related to the targeted special issue. The paper has a sound structure but there are some serious issues which need authors' attention. They are listed below:

1. Introduction: The rainfall erosivity in RUSLE is well defined and widely used. The authors have also used as benchmark to assess their empirical relationship developed. It is necessary to explain why new empirical relationship needs to be developed. What are the main issues prevent its use in the area. 

2. Representativeness of the rainfall period: the rainfall period (2011 to 2017) is relatively short. Please justify its representativeness by compare it to longer time period data. It will also help to define so called wet / normal / dry year mentioned in the paper.

3. Standard of erosive rainfall:  it is not clear how the so called standard erosive rainfall was selected as there is not much erosion data presented. Since it is a key concept. More details are required.

4. Assessment of the empirical relationships for rainfall erosiovity: the derived rainfall erosivity from rainfall-based, composite-based, RUSLE-based approaches should all be related to the monitored erosion modules and comparison should be made to judge which performs best. This could be used to support your argument on the development of a new R factor. 

5. Wider implications: Authors have to discuss the wider significance of the work. As it stands, it is a local study and the new relationships can not be applied anywhere else. The potential impacts of projected climate changes could also be mentioned

6. Limitations of the work: I think authors should acknowledge any limitations with their work, including shorter rainfall time series. 

English writing definitely needs improvement. I have attached an annotated version of the paper where problematic areas were highlighted and more detailed suggestions on figures and tables were also made for your consideration.

Author Response

Dear Reviewer,

This paper has presented the detailed analysis of rainfall characteristics in a study site in China where plot-based monitoring results were used to identify threshold value for erosive rainfall, quantification of their temporal and spatial variations and develop customised methodology for the estimation of rainfall erosivity. The topic covered (water and soil management on agricultural land) is related to the targeted special issue. The paper has a sound structure but there are some serious issues which need authors' attention. They are listed below:

Point 1: Introduction: The rainfall erosivity in RUSLE is well defined and widely used. The authors have also used as benchmark to assess their empirical relationship developed. It is necessary to explain why new empirical relationship needs to be developed. What are the main issues prevent its use in the area.

Response 1: The rainfall erosivity is EI30 in RUSLE. It is complicated to calculate EI30 and rainfall process data are needed. So, many simple methods of estimating rainfall erosivity have been studied, and rainfall erosivity has been evaluated and calculated using the conventional rainfall statistical data of meteorological stations. We have explained and gave an example for its development and the main issues prevent its use in the introduction (L45-53).

In addition, we have supplemented the reference of [11] for estimating rainfall erosivity in previous line 50.

Point 2: Representativeness of the rainfall period: the rainfall period (2011 to 2017) is relatively short. Please justify its representativeness by compare it to longer time period data. It will also help to define so called wet / normal / dry year mentioned in the paper.

Response 2: We have supplemented some examples of the rainfall period for calculating rainfall erosivity. Their rainfall periods were from 3 to 16 years, including 7 years the same as ours (L224-228). The precipitation of Dongtai City in wet years, normal years, and dry years were added from comprehensive planning of water resources in Dongtai City. Representativeness of the rainfall period (2011 to 2017) was explained in previous lines 227-233.

Point 3: Standard of erosive rainfall:  it is not clear how the so called standard erosive rainfall was selected as there is not much erosion data presented. Since it is a key concept. More details are required.

Response 3: The steps in the calculation process of erosive rainfall standard were supplemented in previous lines 203-210. We found that the We found that the words of “264 rainfall events” in previous line 199 should be “258 rainfall events”.

Point 4: Assessment of the empirical relationships for rainfall erosiovity: the derived rainfall erosivity from rainfall-based, composite-based, RUSLE-based approaches should all be related to the monitored erosion modules and comparison should be made to judge which performs best. This could be used to support your argument on the development of a new R factor.

Response 4: According to comparison of rainfall-based, composite-based, RUSLE-based approaches, we got the recommended empirical formulas under different conditions: the composite model is recommended for calculating the single and monthly rainfall erosivity, and the rainfall-based model can be used for calculating the annual rainfall erosivity in Jiangsu coastal areas, and added their basis to the discussion (L411-418) . At the same time, similar parts were removed from the conclusion (L459-464).

Point 5: Wider implications: Authors have to discuss the wider significance of the work. As it stands, it is a local study and the new relationships can not be applied anywhere else. The potential impacts of projected climate changes could also be mentioned.

Response 5: This work was based on the observed data in Jiangsu coastal areas. The property of the areas was stressed in the introduction (L85-86). The application scope of the new relationships was defined in the summary, discussion and conclusion (L29, L417-418, L468). The influence of climate change on empirical formulas were added in the introduction and discussion (L53-55, L419-422).

In addition, we have supplemented the reference of [3] for the areas in previous line 86.

Point 6: Limitations of the work: I think authors should acknowledge any limitations with their work, including shorter rainfall time series.

Response 6: This work was based on regional data and statistical analysis and limited to expend their application. As shown in the responses to Point 2 and Point 5, this study should be used in Jiangsu coastal areas, and need to be tested and modified by using more recent data to improve their accuracy and stability (L29-30, L53-55, L419-422, L466-468). In the follow-up experimental process, we will strengthen the research in this area.

Comments on the Quality of English Language: English writing definitely needs improvement. I have attached an annotated version of the paper where problematic areas were highlighted and more detailed suggestions on figures and tables were also made for your consideration.

Response: This manuscript was previously submitted to MDPI for English editing with the following certificate.

This paper has presented the detailed analysis of rainfall characteristics in a study site in China where plot-based monitoring results were used to identify threshold value for erosive rainfall, quantification of their temporal and spatial variations and develop customised methodology for the estimation of rainfall erosivity. The topic covered (water and soil management on agricultural land) is related to the targeted special issue. The paper has a sound structure but there are some serious issues which need authors' attention. They are listed below:

Point 1: Introduction: The rainfall erosivity in RUSLE is well defined and widely used. The authors have also used as benchmark to assess their empirical relationship developed. It is necessary to explain why new empirical relationship needs to be developed. What are the main issues prevent its use in the area.

Response 1: The rainfall erosivity is EI30 in RUSLE. It is complicated to calculate EI30 and rainfall process data are needed. So, many simple methods of estimating rainfall erosivity have been studied, and rainfall erosivity has been evaluated and calculated using the conventional rainfall statistical data of meteorological stations. We have explained and gave an example for its development and the main issues prevent its use in the introduction (L45-53).

In addition, we have supplemented the reference of [11] for estimating rainfall erosivity in previous line 50.

Point 2: Representativeness of the rainfall period: the rainfall period (2011 to 2017) is relatively short. Please justify its representativeness by compare it to longer time period data. It will also help to define so called wet / normal / dry year mentioned in the paper.

Response 2: We have supplemented some examples of the rainfall period for calculating rainfall erosivity. Their rainfall periods were from 3 to 16 years, including 7 years the same as ours (L224-228). The precipitation of Dongtai City in wet years, normal years, and dry years were added from comprehensive planning of water resources in Dongtai City. Representativeness of the rainfall period (2011 to 2017) was explained in previous lines 227-233.

Point 3: Standard of erosive rainfall:  it is not clear how the so called standard erosive rainfall was selected as there is not much erosion data presented. Since it is a key concept. More details are required.

Response 3: The steps in the calculation process of erosive rainfall standard were supplemented in previous lines 203-210. We found that the We found that the words of “264 rainfall events” in previous line 199 should be “258 rainfall events”.

Point 4: Assessment of the empirical relationships for rainfall erosiovity: the derived rainfall erosivity from rainfall-based, composite-based, RUSLE-based approaches should all be related to the monitored erosion modules and comparison should be made to judge which performs best. This could be used to support your argument on the development of a new R factor.

Response 4: According to comparison of rainfall-based, composite-based, RUSLE-based approaches, we got the recommended empirical formulas under different conditions: the composite model is recommended for calculating the single and monthly rainfall erosivity, and the rainfall-based model can be used for calculating the annual rainfall erosivity in Jiangsu coastal areas, and added their basis to the discussion (L411-418) . At the same time, similar parts were removed from the conclusion (L459-464).

Point 5: Wider implications: Authors have to discuss the wider significance of the work. As it stands, it is a local study and the new relationships can not be applied anywhere else. The potential impacts of projected climate changes could also be mentioned.

Response 5: This work was based on the observed data in Jiangsu coastal areas. The property of the areas was stressed in the introduction (L85-86). The application scope of the new relationships was defined in the summary, discussion and conclusion (L29, L417-418, L468). The influence of climate change on empirical formulas were added in the introduction and discussion (L53-55, L419-422).

In addition, we have supplemented the reference of [3] for the areas in previous line 86.

Point 6: Limitations of the work: I think authors should acknowledge any limitations with their work, including shorter rainfall time series.

Response 6: This work was based on regional data and statistical analysis and limited to expend their application. As shown in the responses to Point 2 and Point 5, this study should be used in Jiangsu coastal areas, and need to be tested and modified by using more recent data to improve their accuracy and stability (L29-30, L53-55, L419-422, L466-468). In the follow-up experimental process, we will strengthen the research in this area.

Comments on the Quality of English Language: English writing definitely needs improvement. I have attached an annotated version of the paper where problematic areas were highlighted and more detailed suggestions on figures and tables were also made for your consideration.

Response: This manuscript was previously submitted to MDPI for English editing with the following certificate.

Round 2

Reviewer 3 Report

Authors have made improvements to the paper but I do not agree with some of the arguments provided.

1) Introduction: the authors have not made good case for the development of a local empirical relationships. There are relatively high resolution mapped erosivity available (e.g. https://www.nature.com/articles/s41598-017-04282-8). Authors have not reference relevant datasets and clarified why the existing data could not be used. The argument of 'too complicate to calculate' is very weak. The vague 'rainfall process data' does not help either.

2) Rainfall record: the authors have provided some existing papers to show similar length of data record has been used in the past. Every data record is unique. Climate related studied usually have data records between 20 to 30 years long. For shorter data record, it is necessary for authors to demonstrate that their data is representative and sufficient for the intended studies.

3) Definition of erosive rainfall standard: I have misunderstood your approach. My apologise here. I must say that the description of the procedure is not easy to comprehend and I am surprised to find out that only rainfall data was used and no reference to monitored flow and soil loss data at all. 

4) Clearly, you have not received my annotated version of the paper where I have highlighted some issues and raised more questions. I have attached it with this review form for your reference only. 

While authors' have provided evidence that the paper has gone through MDPI editing process, I would argue that there are still some issues. 

Author Response

Dear Reviewer,

Authors have made improvements to the paper but I do not agree with some of the arguments provided.

1) Introduction: the authors have not made good case for the development of a local empirical relationships. There are relatively high resolution mapped erosivity available (e.g. https://www.nature.com/articles/s41598-017-04282-8). Authors have not reference relevant datasets and clarified why the existing data could not be used. The argument of too complicate to calculate' is very weak. The vague 'rainfall process data' does not help either.

2) Rainfall record: the authors have provided some existing papers to show similar length of data record has been used in the past. Every data record is unique. Climate related studied usually have data records between 20 to 30 years long. For shorter data record, it is necessary for authors to demonstrate that their data is representative and sufficient for the intended studies.

3) Definition of erosive rainfall standard: I have misunderstood your approach. My apologise here. I must say that the description of the procedure is not easy to comprehend and I am surprised to find out that only rainfall data was used and no reference to monitored flow and soil loss data at all.

4) Clearly, you have not received my annotated version of the paper where I have highlighted some issues and raised more questions. I have attached it with this review form for your reference only.

They are listed below:

Point 1: Introduction: the authors have not made good case for the development of a local empirical relationships. There are relatively high resolution mapped erosivity available (e.g. https://www.nature.com/articles/s41598-017-04282-8). Authors have not reference relevant datasets and clarified why the existing data could not be used. The argument of ‘too complicate to calculate' is very weak. The vague 'rainfall process data' does not help either.

Response 1: We are grateful for the comment. Panagos et al.(2017) presented the results of an extensive global data collection effort including the stations (located in China) the calculation of the rainfall erosivity, based on high temporal resolution data (1998-2012). Moreover, for Jiangsu coastal areas’ special environment characteristics (synoptic climate over the offshore, newly formed land reclaimed and silted in coastal areas), we preferred to get local empirical relationships based on the observed and rainfall data, since it’s the way adapting it to local context. We have supplemented the explanation in previous lines 66-70, and 97-99.

In addition, we have supplemented the reference of [12] for the areas in previous line 66.

Point 2: Rainfall record: the authors have provided some existing papers to show similar length of data record has been used in the past. Every data record is unique. Climate related studied usually have data records between 20 to 30 years long. For shorter data record, it is necessary for authors to demonstrate that their data is representative and sufficient for the intended studies.

Response 2: The comparison between past 30 years (1970-2000) and 14 years (1998-2012, used in the Global Rainfall Erosivity Database) were added to demonstrate that our data are representative and sufficient for the studies, and the bias due to periods for calculating rainfall erosivity should be considered. Furthermore, the period for calculating rainfall erosivity was the same as the period of observed soil loss data. We have supplemented the comparison and advice in previous lines231-233, and 462-463.

Point 3: Clearly, you have not received my annotated version of the paper where I have highlighted some issues and raised more questions. I have attached it with this review form for your reference only.

Response 3: We have revised some expressions and writing errors in the manuscript according to your review. The revisions are included in the manuscript.

If you have any questions about this manuscript, please don’t hesitate to contact us.

Thanks again for your valuable comments and suggestions.

Best wishes

Feng Chen 
